# Development of an amplicon-based sequencing approach in response to the global emergence of mpox

**Nicholas F. G. Chen[1‡], Chrispin Chaguza[1‡], Luc Gagne[2‡], Matthew Doucette[2], Sandra Smole[2], Erika Buzby[2], Joshua Hall[2], Stephanie Ash[2], Rachel Harrington[2], Seana Cofsky[2], Selina Clancy[2], Curtis J. Kapsak[3], Joel Sevinsky[3], Kevin Libuit[3], Daniel J. Park[4], Peera Hemarajata[5], Jacob M. Garrigues[5], Nicole M. Green[5], Sean Sierra-Patev[6], Kristin Carpenter-Azevedo[6], Richard C. Huard[6], Claire Pearson[7], Kutluhan Incekara[7], Christina Nishimura[7], Jian Ping Huang[7], Emily Gagnon[7], Ethan Reever[7], Jafar Razeq[7], Anthony Muyombwe[7], Vítor Borges[8], Rita Ferreira[8], Daniel Sobral[8], Silvia Duarte[9], Daniela Santos[9], Luís Vieira[9], João Paulo Gomes[8,10], Carly Aquino[11], Isabella M. Savino[11], Karinda Felton[11], Moneeb Bajwa[11], Nyjil Hayward[11], Holly Miller[11], Allison Naumann[11], Ria Allman[11], Neel Greer[11], Amary Fall[12], Heba H. Mostafa[12], Martin P. McHugh[13,14], Daniel M. Maloney[13,15], Rebecca Dewar[13], Juliet Kenicer[13], Abby Parker[13], Katharine Mathers[13], Jonathan Wild[13], Seb Cotton[13], Kate E. Templeton[13], George Churchwell[16], Philip A. Lee[16], Maria Pedrosa[16], Brenna McGruder[16], Sarah Schmedes[16], Matthew R. Plumb[17], Xiong Wang[17], Regina Bones Barcellos[18], Fernanda M. S. Godinho[18], Richard Steiner Salvato[18], Aimee Ceniseros[19], Mallery I. Breban[1], Nathan D. Grubaugh[1,20], Glen R. Gallagher[2,6‡]\*, Chantal B. F. Vogels[1‡]\***

1 Department of Epidemiology of Microbial Diseases, Yale School of Public Health, New Haven, Connecticut, United States of America, 2 Massachusetts Department of Public Health, Jamaica Plain, Massachusetts, United States of America, 3 Theiagen Genomics, Highlands Ranch, Colorado, United States of America, 4 Broad Institute, Cambridge, Massachusetts, United States of America, 5 Los Angeles County Public Health Laboratories, Downey, California, United States of America, 6 Rhode Island Department of Health, Rhode Island State Health Laboratory, Providence, Rhode Island, United States of America, 7 Connecticut Department of Public Health, Rocky Hill, Connecticut, United States of America, 8 Genomics and Bioinformatics Unit, Department of Infectious Diseases, National Institute of Health Doutor Ricardo Jorge (INSA), Lisbon, Portugal, 9 Technology and Innovation Unit, Department of Human Genetics, National Institute of Health Doutor Ricardo Jorge (INSA), Lisbon, Portugal, 10 Faculty of Veterinary Medicine, Lusófona University, Lisbon, Portugal, 11 Delaware Public Health Laboratory, Smyrna, Delaware, United States of America, 12 Johns Hopkins School of Medicine, Baltimore, Maryland, United States of America, 13 Viral Genotyping Reference Laboratory Edinburgh, NHS Lothian, Royal Infirmary of Edinburgh, Edinburgh, United Kingdom, 14 School of Medicine, University of St Andrews, St Andrews, United Kingdom, 15 Institute of Ecology and Evolution, University of Edinburgh, Edinburgh, United Kingdom, 16 Florida Department of Health, Bureau of Public Health Laboratories, Jacksonville, Florida, United States of America, 17 Minnesota Department of Health, Public Health Laboratory, St. Paul, Minnesota, United States of America, 18 Centro Estadual de Vigilância em Saúde, Secretaria Estadual da Saúde do Rio Grande do Sul, Porto Alegre, Rio Grande do Sul, Brazil, 19 Idaho Bureau of Laboratories, Boise, Idaho, United States of America, 20 Department of Ecology and Evolutionary Biology, Yale University, New Haven, Connecticut, United States of America

‡ NFGC, CC, and LG share first authorship on this work. GRG and CBFV are co-senior authors on this work.
\* glen.gallagher@health.ri.gov (GRG); chantal.vogels@yale.edu (CBFV)

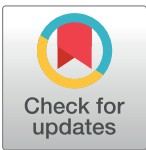

**Data Availability Statement:** Genomic data are available on NCBI Sequence Read Archive (SRA), GenBank, and GISAID (see accession numbers in

## Abstract

The 2022 multicountry mpox outbreak concurrent with the ongoing Coronavirus Disease 2019 (COVID-19) pandemic further highlighted the need for genomic surveillance and rapid pathogen whole-genome sequencing. While metagenomic sequencing approaches have

S3 Table). All other data are included in the manuscript and the Supporting Information.

**Funding:** This publication was made possible by CTSA Grant Number UL1 TR001863 from the National Center for Advancing Translational Science (NCATS), a component of the National Institutes of Health (NIH) awarded to CBFV. INSA was partially funded by the HERA project (Grant/ 2021/PHF/23776) supported by the European Commission through the European Centre for Disease Control (to VB). The funders had no role in study design, data collection and analysis, decision to publish, or preparation of the manuscript.

**Competing interests:** I have read the journal's policy and the authors of this manuscript have the following competing interests: NDG is a consultant for Tempus Labs and the National Basketball Association for work related to COVID-19. All other authors have declared that no competing interests exist.

**Abbreviations:** CDPH, Connecticut Department of Public Health; CEVS, Centro Estadual de Vigilância em Saúde; COVID-19, Coronavirus Disease 2019; Ct, cycle threshold; DPHL, Delaware Public Health Lab; FDA, US Food and Drug Administration; FDH, Florida Department of Health; IBL, Idaho Bureau of Laboratories; INSA, National Institute of Health Dr. Ricardo Jorge; JHMI, Johns Hopkins Medical Institutions; LACPHL, Los Angeles County Public Health Lab; LMIC, low- or middle-income country; MASPHL, Massachusetts State Public Health Laboratory; MDH, Minnesota Department of Health; NHS Lothian, National Health Service Lothian; RIDOH RISHL, Rhode Island Department of Health/Rhode Island State Health Laboratory; SARS-CoV-2, Severe Acute Respiratory Syndrome Coronavirus 2; YSPH, Yale School of Public Health.

been used to sequence many of the early mpox infections, these methods are resource intensive and require samples with high viral DNA concentrations. Given the atypical clinical presentation of cases associated with the outbreak and uncertainty regarding viral load across both the course of infection and anatomical body sites, there was an urgent need for a more sensitive and broadly applicable sequencing approach. Highly multiplexed amplicon-based sequencing (PrimalSeq) was initially developed for sequencing of Zika virus, and later adapted as the main sequencing approach for Severe Acute Respiratory Syndrome Coronavirus 2 (SARS-CoV-2). Here, we used PrimalScheme to develop a primer scheme for human monkeypox virus that can be used with many sequencing and bioinformatics pipelines implemented in public health laboratories during the COVID-19 pandemic. We sequenced clinical specimens that tested presumptively positive for human monkeypox virus with amplicon-based and metagenomic sequencing approaches. We found notably higher genome coverage across the virus genome, with minimal amplicon drop-outs, in using the amplicon-based sequencing approach, particularly in higher PCR cycle threshold (Ct) (lower DNA titer) samples. Further testing demonstrated that Ct value correlated with the number of sequencing reads and influenced the percent genome coverage. To maximize genome coverage when resources are limited, we recommend selecting samples with a PCR Ct below 31 Ct and generating 1 million sequencing reads per sample. To support national and international public health genomic surveillance efforts, we sent out primer pool aliquots to 10 laboratories across the United States, United Kingdom, Brazil, and Portugal. These public health laboratories successfully implemented the human monkeypox virus primer scheme in various amplicon sequencing workflows and with different sample types across a range of Ct values. Thus, we show that amplicon-based sequencing can provide a rapidly deployable, cost-effective, and flexible approach to pathogen whole-genome sequencing in response to newly emerging pathogens. Importantly, through the implementation of our primer scheme into existing SARS-CoV-2 workflows and across a range of sample types and sequencing platforms, we further demonstrate the potential of this approach for rapid outbreak response.

## Introduction

The integration of pathogen whole-genome sequencing with public health surveillance provides a powerful tool to inform outbreak control [1,2]. While the feasibility of real-time genomic surveillance was demonstrated during the 2013 to 2016 Ebola outbreak [3], the Severe Acute Respiratory Syndrome Coronavirus 2 (SARS-CoV-2) pandemic has launched a revolution in viral genomics [4]. To date, more than 14 million SARS-CoV-2 genomes have been sequenced and shared publicly [5], furthering our understanding of viral transmission and evolution. The rapid advancement in pathogen genomics in public health laboratories was facilitated in the United States through significant investments by the Centers of Disease Control and Prevention's Office of Advanced Molecular Detection. These programs situated sequencing equipment in state and local public health laboratories and provided practical training in laboratory and bioinformatics approaches to allow for the rapid adoption of new sequencing methods. As the Coronavirus Disease 2019 (COVID-19) pandemic remains ongoing, the recent spread of human monkeypox virus outside of endemic areas has provided a new target for genomic surveillance efforts [6].

Monkeypox is a zoonotic DNA virus of the *Orthopox* genus endemic to Western and Central Africa [7]. The virus consists of 3 major clades (I, IIa, and IIb), with a subgroup of clade IIb being referred to as human monkeypox virus due to direct transmission from human to human [8]. Initially rare outside endemic countries beyond imported cases, mpox has emerged as a global threat. It was first detected in the United Kingdom on May 7, 2022, quickly spreading to other continents through travel-related infections and sometimes unknown transmission chains [9]. As of January 5, 2023, 84,075 cases of mpox across 103 non-endemic countries have been reported, often with atypical clinical presentations [6,10]. The lack of consistent clinical presentations, unknown transmission dynamics, and uncertainty surrounding animal reservoirs highlights the importance of establishing rapid genomic surveillance networks.

Much of the early human monkeypox virus sequencing during the 2022 global outbreak has been accomplished via a metagenomics approach [9], which employs sequencing of the total nucleic acid present in a sample. While considered the gold standard for untargeted sequencing, metagenomics incurs a high resource cost, requires experienced sequencing and bioinformatics teams, is computationally demanding, and relies on samples with high viral concentrations relative to background nucleic acids [11]. The initial metagenomic approaches included hybrid assemblies of both long and short read sequencing to allow for high confidence and polishing of the early genomes. Although this was important to establish reference genomes, it comes at the cost of being able to sequence larger numbers of specimens [12]. These characteristics of metagenomic sequencing make the approach less suitable for rapid response to large-scale pathogen outbreaks, which require low-cost, rapidly deployable solutions able to be implemented across a range of settings, sample types, and experience levels.

Highly multiplexed amplicon-based sequencing (PrimalSeq) was initially developed to generate greater coverage and depth for Zika virus genomes [13] and was later adapted as the main sequencing approach during the SARS-CoV-2 pandemic for both Illumina and Oxford Nanopore Technologies sequencing platforms [14]. Here, we developed a primer scheme for human monkeypox virus using PrimalScheme for use with amplicon-based sequencing workflows widely established during the COVID-19 pandemic. We show a consistently high percent or complete genome coverage across a range of PCR cycle threshold (Ct) values, with different workflows and sequencing platforms. Our amplicon-based sequencing approach provides a more sensitive, lower cost, and higher throughput strategy that can be "plugged" into currently established genomic infrastructure, providing an invaluable tool for public health surveillance of human monkeypox virus.

## Results

In May 2022, a growing cluster of mpox cases in humans was reported outside its endemic region [6,10]. Difficulties in obtaining sufficient coverage with metagenomic sequencing approaches led us to develop a primer scheme for use with amplicon-based sequencing approaches. Given that many of the early B.1 outbreak clade genomes had low coverage, we used the closely related pre-outbreak A.1 clade genome (GenBank accession: MT903345) as a reference for the primer scheme. The primer scheme, designed using PrimalScheme [15], consists of 163 primer pairs with an amplicon length ranging between 1,597 and 2,497 bp (average length of 1,977 bp; **S1 Table**). For the initial validation, we sequenced 10 clinical specimens with a range of PCR Ct values with both amplicon-based and metagenomic sequencing approaches at the Massachusetts State Public Health Laboratory (MASPHL). Clinical specimens ranged in Ct value from 15.0 (highest DNA concentration) to 34.6 (lowest DNA concentration). We found comparable genome coverage between amplicon and metagenomic sequencing with low Ct (<18) samples, but dramatically higher genome coverage with

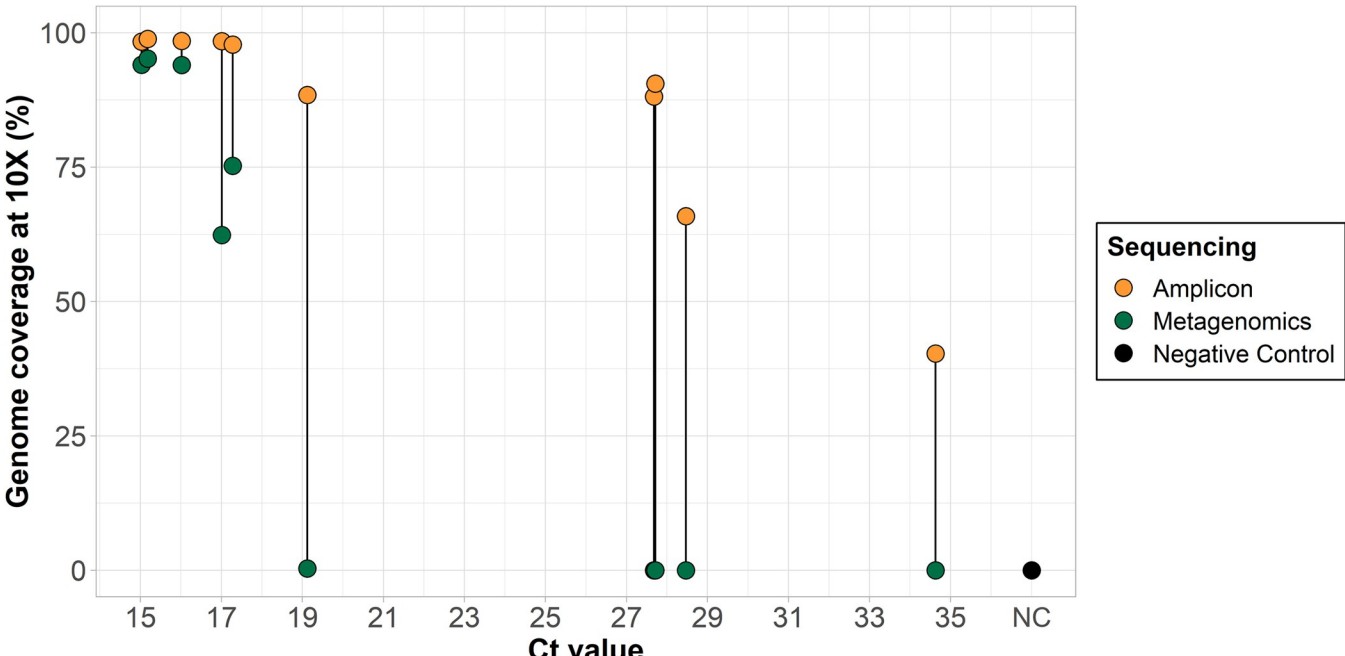

**Fig 1. Comparison of percent genome coverage at 10× of clinical specimens sequenced with amplicon-based and metagenomic sequencing approaches.** DNA was extracted from 10 clinical samples manually extracted with the QIAamp DSP DNA Blood Mini kit and PCR Ct values were determined with the non-variola *Orthopox* real-time PCR assay. Libraries were prepared with amplicon-based and metagenomic sequencing approaches and sequenced on the Illumina MiSeq (2 × 150 bp) with a targeted 0.5–1 million reads per library for amplicon-based sequencing and 1.5–3 million reads per library for metagenomic sequencing. A negative template control was included during library prep for each sequencing run. For amplicon-based sequencing, consensus genomes were generated at a read depth coverage of 10× and percent genome coverage as compared to the reference genome (MT903345) was determined using TheiaCoV_Illumina_PE Workflow Series on Terra.bio. For metagenomic sequencing, genomes were generated using the Broad Institute's viral-pipelines workflows on Terra.bio using both the assemble_refbased and assemble_denovo workflows. Source data can be found in S1 Data. Ct, cycle threshold.

amplicon sequencing in higher Ct samples (>18; **Fig 1 and S1 Data**) as compared to metagenomics sequencing. These initial findings show that amplicon-based sequencing approaches can help to improve genome coverage of human monkeypox virus genomes from clinical specimens, particularly at higher Ct values.

To further test the primer scheme, we sequenced an additional 145 clinical specimens in 2 independent laboratories (**Fig 2 and S2 Data**). These samples consisted of 115 lesion swabs and 8 oropharyngeal swabs sequenced at the MASPHL, and 22 lesion swabs collected by the Connecticut Department of Public Health (CDPH) and sequenced at the Yale School of Public Health (YSPH). Our sequencing results from both laboratories confirmed that we were able to successfully sequence human monkeypox virus from lesion swabs with a range of Ct values using 2 different library prep kits (Illumina DNA prep (MASPHL) and Illumina COVIDSeq test (CDPH/YSPH) kits). Lesion swabs were the most common sample type used, as this sample type is recommended by the US Food and Drug Administration (FDA) for clinical diagnostics [16]. However, several studies have shown that human monkeypox virus can also be detected in other sample types such as throat or oropharyngeal swabs, saliva, feces, urine, and semen [17,18]. MASPHL had access to alternate sample types and sequenced 8 oropharyngeal swabs with or without the presence of lesions in the throat. Although the total number of samples is low, we show that near-complete genomes can be sequenced from oropharyngeal swabs, in the presence and absence of lesions (**Fig 2A**). This shows that both lesion and oropharyngeal swabs could serve as sample types for sequencing.

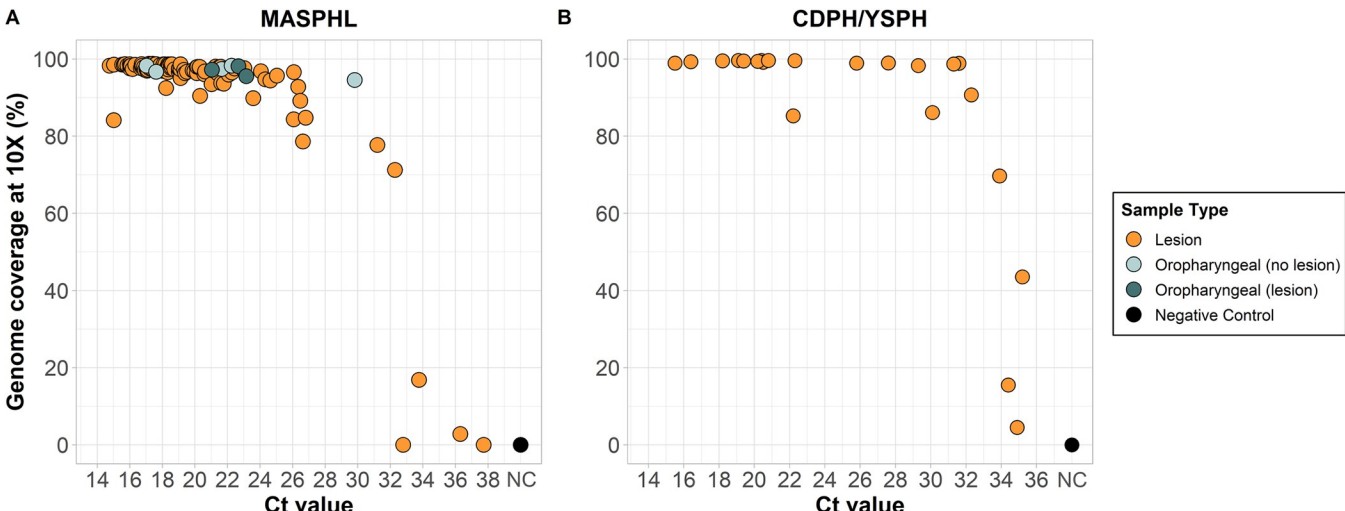

**Fig 2. Percent genome coverage at 10× for clinical specimens sequenced with the amplicon-based sequencing approach.** (**A**) Clinical specimens ($n$ = 123) consisting of 115 lesion swabs, 5 oropharyngeal swabs in the absence of lesions, and 3 oropharyngeal swabs in the presence of lesions sequenced by the MASPHL. Libraries were prepared using the Illumina DNA prep kit and sequenced on the MiSeq with 0.5–1 million reads per sample. A negative template control was included during library prep for each sequencing run. (**B**) Lesion swabs ($n$ = 22) obtained from 12 individuals through the CDPH and sequenced by the YSPH. Libraries were prepared using the Illumina COVIDSeq test (RUO version) and sequenced on the NovaSeq with on average 12 million reads per sample. A negative template control was included during library prep. Bioinformatic analyses were unified between both laboratories using iVar with a minimum read depth of 10. Source data can be found in S2 Data. CDPH, Connecticut Department of Public Health; Ct, cycle threshold; MASPHL, Massachusetts State Public Health Laboratory; YSPH, Yale School of Public Health.

While both laboratories utilized the same primer scheme, there was a greater variation in percent genome coverage at lower Ct values in the MASPHL data compared to CDPH/YSPH. This difference in coverage between the 2 laboratories may be due to the difference in sequencing reads allotted to each sample. MASPHL sequenced on the Illumina MiSeq with 0.5 to 1 million reads/sample, whereas CDPH/YSPH sequenced on the Illumina NovaSeq which generated a higher output resulting in on average approximately 12 million reads per sample (range: approximately 0.4 to approximately 20 million reads). To further investigate the optimal number of sequencing reads per sample, we randomly down-sampled the 22 clinical specimens sequenced by CDPH/YSPH to 0.5, 1, 1.5, and 2 million sequencing reads per sample. We chose data from CDPH/YSPH for down-sampling because the large number of initial reads per sample allowed for a wider range of down-sampling read depths than would be possible with the data from MASPHL. To better understand the threshold for sequencing, we then used a logistic function analysis to determine the PCR Ct value threshold to achieve 80% genome coverage at 10× read depth (i.e., at least 10 sequencing reads aligned to a genome position) for each level of down-sampling. We found that down-sampling the total number of sequencing reads from 2 million to 0.5 million decreased the 80% coverage Ct threshold by 1 Ct (from 32.3 Ct to 31.3 Ct) and resulted in an average 7.2% (range: 0% to 26.1%) decrease in percent genome coverage at 10× (**Fig 3 and S3 Data**). To achieve an overall high genome coverage of more than 80% at 10× read depth, we recommend generating at least 1 million sequencing reads per sample, if resources allow. Further increasing the number of sequencing reads to 2 million helps to generate higher coverage for samples with relatively high Ct values >31, but only small differences in coverage at 10× read depth of on average 2.1% were observed with samples that had Ct values <31. Thus, we recommend selecting samples for sequencing with a Ct value below 31 and generating at least 1 million reads per sample, to maximize genome coverage, particularly when resources are limited.

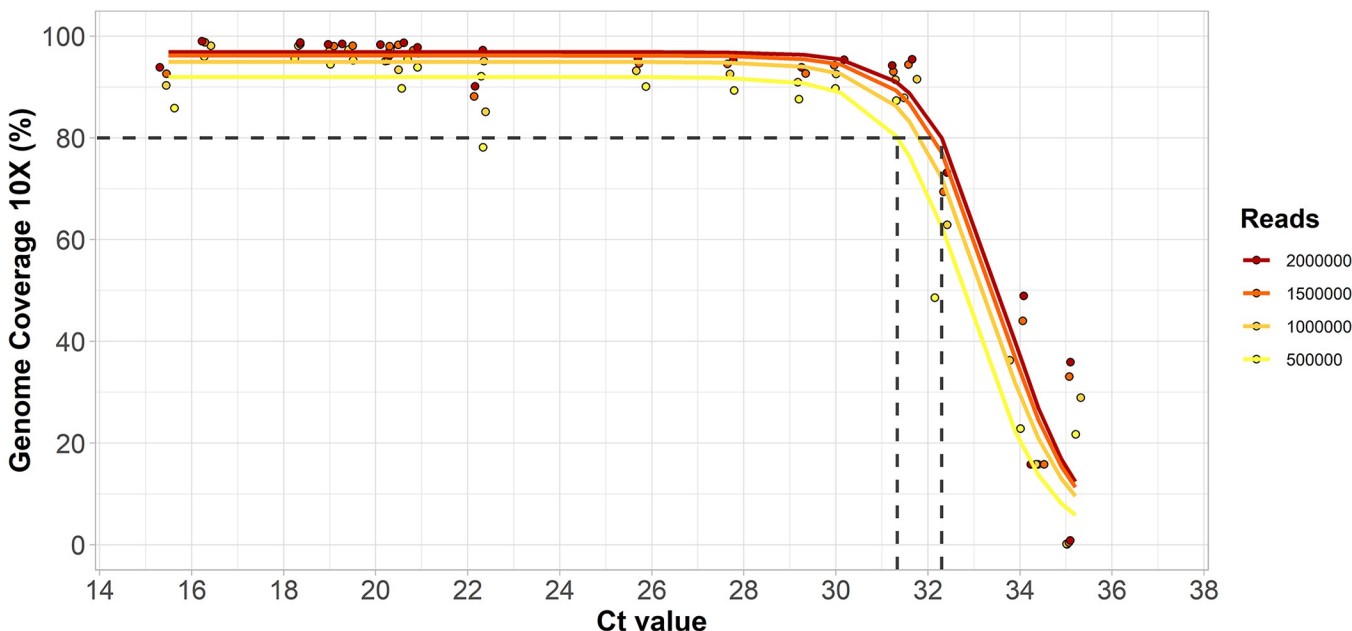

**Fig 3. Percent genome coverage at 10× mapped read depth for 22 clinical specimens after randomly down-sampling to a specific number of sequencing reads.** To further investigate the combined effects of Ct value and number of sequencing reads per sample, we randomly down-sampled the CDPH/YSPH data to 2, 1.5, 1, and 0.5 million reads per sample. We used a logistic function analysis to plot the fitted lines indicating the decrease in percent genome coverage with higher Ct values. Source data can be found in S3 Data. CDPH, Connecticut Department of Public Health; Ct, cycle threshold; YSPH, Yale School of Public Health.

To investigate the differences in coverage depth for the randomly down-sampled data, we determined the depth of coverage at each nucleotide position across the genome. This analysis again confirmed that increasing Ct values and decreasing sequencing reads per sample result in more regions of the genomes with coverage <10× depth (**Fig 4** **and S4 Data**). Moreover, it showed variation in depth of coverage across the genome. To further investigate which specific amplicons consistently had low coverage, we determined the depth of coverage for non-overlapping regions of amplicons in our dataset that was randomly down-sampled to 1 million reads/sample and for samples with a Ct <31. By excluding genomic regions covered by overlapping amplicons, we found that amplicons 75 and 118 consistently had a depth of coverage <10× across all 15 samples included in the analysis. Additionally, the mean depth of coverage for amplicons 11, 26, 28, 56, 59, 60, 74, and 96 was also below 10×. To understand the cause for the lower coverage, we investigated whether there were any mismatches in the primers for these amplicons. This revealed a single mismatch with the MPXV_11_RIGHT primer at the 3′ end, whereas no mismatches were present in any of the other primers. This suggests that differences in PCR efficiency may explain the lower coverage across these amplicons. To further understand how this variation may affect coverage at phylogenetically relevant sites, we compared the coverage of the same 15 samples with Ct <31 and down-sampled to 1 million reads/sample to 46 clade- and lineage-defining mutations [19]. We found that none of the 46 sites had a coverage of <10× across all samples and that only 4 sites, associated with the A.3, B1.1.10, B.1.17, and B.1.5 lineages, had <10× coverage for more than half of the samples (**S2 Table**). Additionally, we show that none of the previously identified amplicons with low mean coverage, with the exception of amplicon 59, are associated with low coverage for these phylogenetically informative sites. These findings suggest that with high-quality samples and a sufficient number of reads per sample, this approach can consistently identify clade- and lineage-

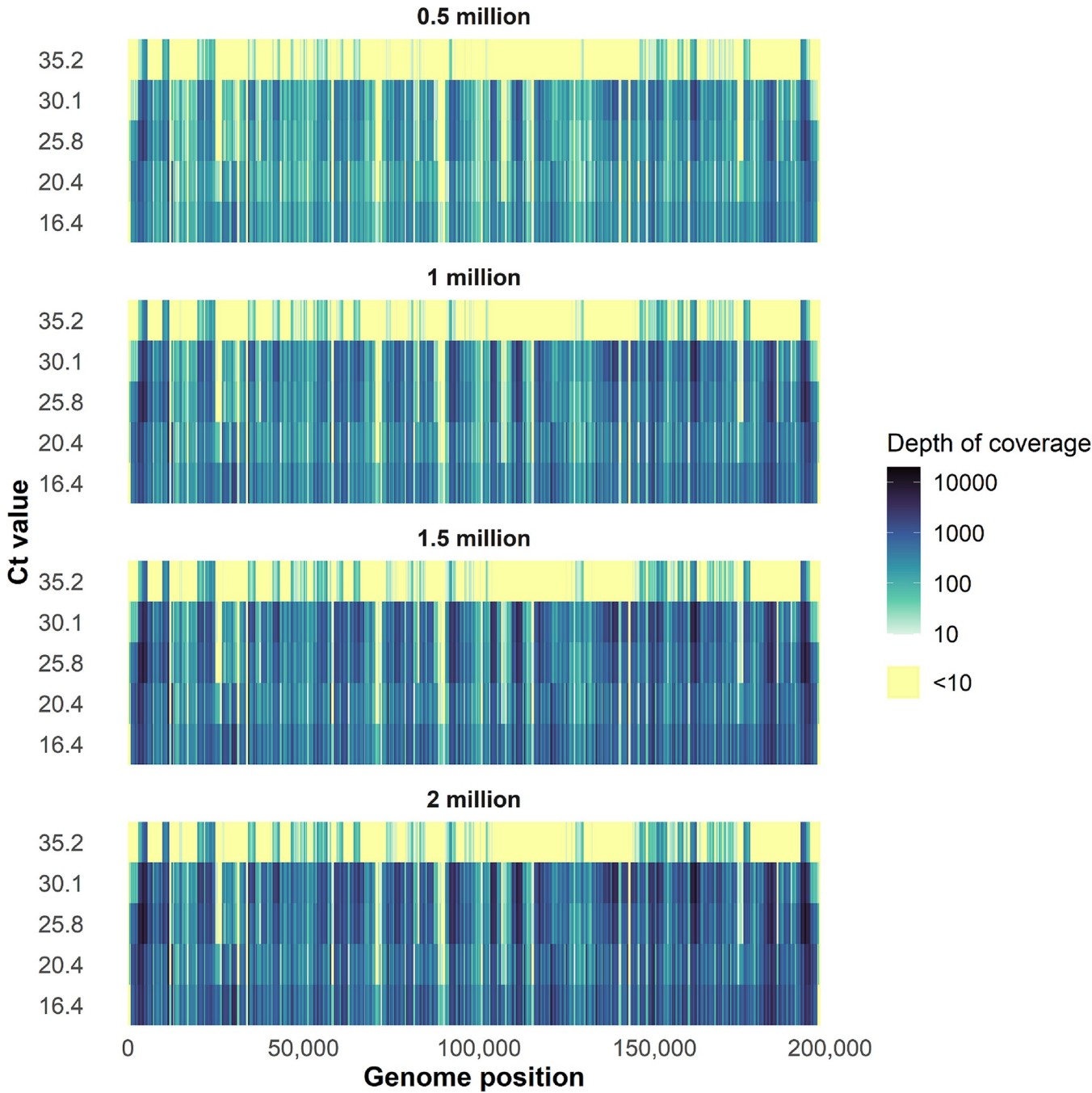

**Fig 4. Depth of coverage by genome position for samples representing a range of Ct values and randomly down-sampled to different numbers of sequencing reads.** We determined the depth of coverage at each nucleotide position for selected samples that represented a range in Ct values from 16.4–35.2, and for which the number of raw sequencing reads was randomly down-sampled to 2, 1.5, 1, and 0.5 million sequencing reads. Each row represents a single specimen, ranked by Ct value from high (low DNA titer) to low (high DNA titer). Highlighted in yellow are positions of the genome with a depth of coverage below 10×. Source data can be found in S4 Data. Ct, cycle threshold.

defining mutations. Further optimization would be required to improve uniformity in PCR efficiency across the primer pairs in the current primer scheme. This may also help to reduce the number of sequencing reads required to reach a minimum coverage of 10× across the entire genome.

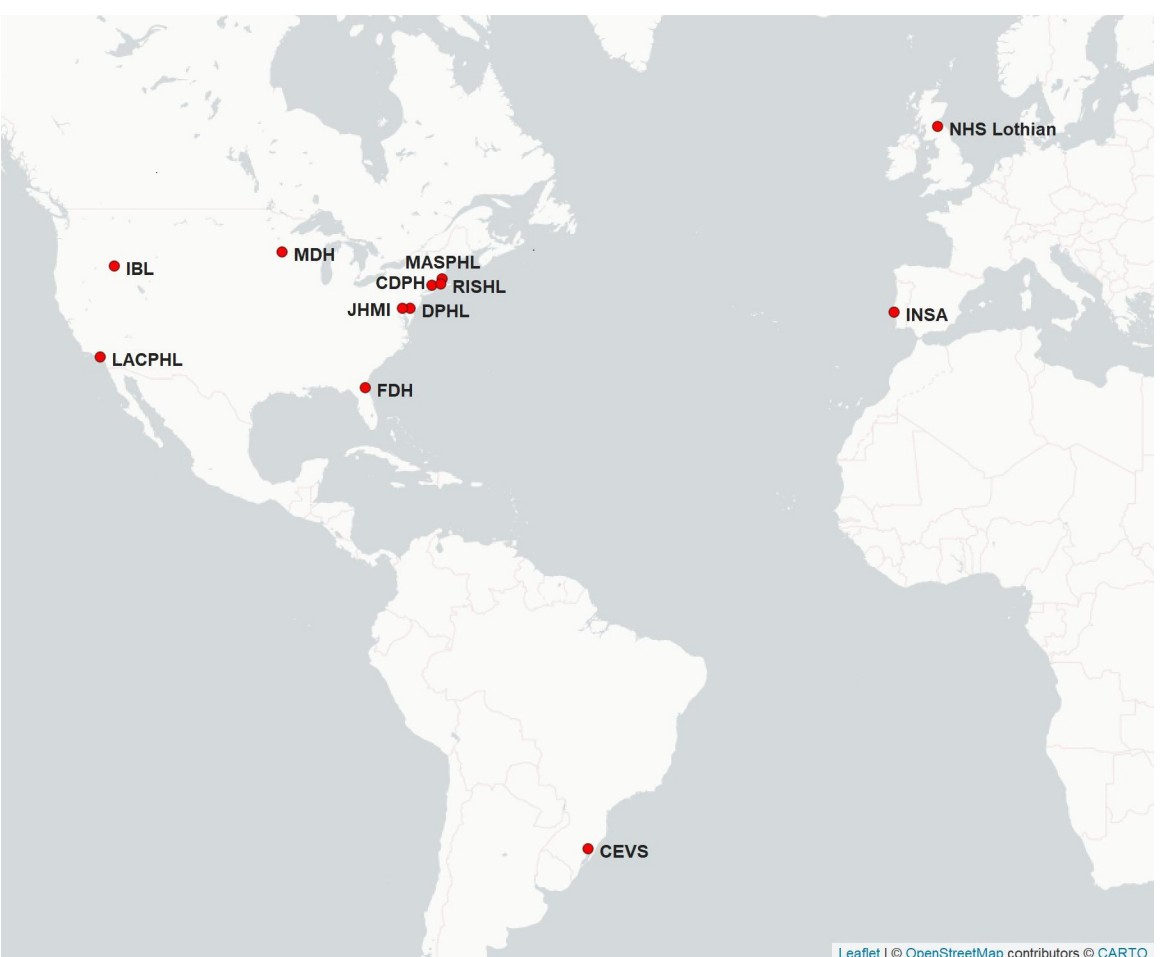

**Fig 5. Geographical distribution of public health laboratories that implemented the human monkeypox virus primer scheme with their established amplicon-based sequencing workflows.** Public health laboratories contributing data to this study include: CDPH, CEVS, DPHL, FDH, IBL, JHMI, LACPHL, MASPHL, MDH, NHS Lothian, INSA, and RISHL. The base layer of the map has been sourced from Carto (https://docs.carto.com/development-tools/carto-for-react/guides/basemaps) under an open source CC-BY license (https://github.com/CartoDB/basemap-styles/blob/master/LICENSE.md). CDPH, Connecticut Department of Public Health; CEVS, Centro Estadual de Vigilância em Saúde; DPHL, Delaware Public Health Lab; FDH, Florida Department of Health; IBL, Idaho Bureau of Laboratories; INSA, National Institute of Health Dr. Ricardo Jorge; JHMI, Johns Hopkins Medical Institutions; LACPHL, Los Angeles County Public Health Lab; MASPHL, Massachusetts State Public Health Laboratory; MDH, Minnesota Department of Health; NHS Lothian, National Health Service Lothian; RISHL, Rhode Island State Health Laboratory.

To support global outbreak response efforts, we immediately made our primer scheme and protocols publicly available following validation [20] and shipped primer pool aliquots to 10 additional public health laboratories across the United States and internationally (**Fig 5**). Each laboratory "plugged" the human monkeypox virus primers into their established SARS-CoV-2 amplicon-based sequencing workflow and sequenced their samples on Illumina or Oxford Nanopore Technologies platforms. We unified the data analysis for all laboratories by running the same bioinformatics pipeline (iVar) to generate consensus genomes for Illumina data and to determine percent genome coverage at 10× read depth. Despite variation between work-flows and sequencing platforms, we found that amplicon-based sequencing resulted in relatively consistent high genome coverage (>80%), with decreasing coverage at higher Ct values (**Fig 6 and S5 Data**). By using a logistic function analysis, we determined the PCR Ct value threshold at which each laboratory was able to achieve 80% genome coverage at 10× read

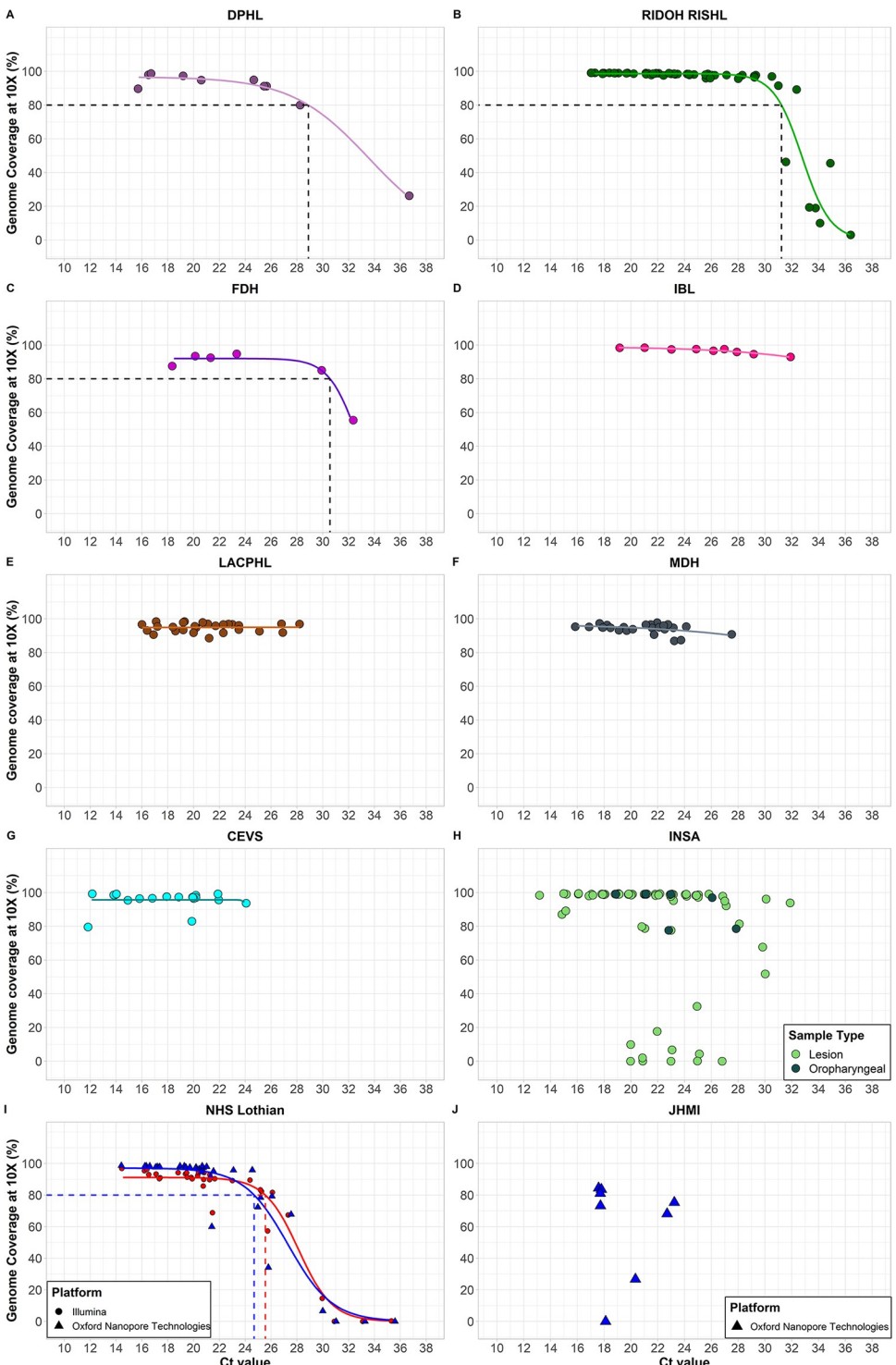

**Fig 6. Percent genome coverage at 10× read depth for clinical specimens sequenced with the amplicon-based sequencing approach.** (**A**) Lesion swabs ($n = 10$) sequenced by the DPHL. Data are fitted with a logistic function and the dashed line corresponds to 80% genome coverage at a threshold of Ct 28.9. (**B**) Dry vesicle swabs ($n = 56$) from various anatomical sites and sequenced by the RIDOH RISHL. Data are fitted with a logistic function and the dashed line corresponds to 80% genome coverage at a threshold of Ct 31.2. (**C**) Lesion swabs ($n = 6$) sequenced by the FDH. Data are fitted with a logistic function and the dashed line corresponds to 80% genome coverage at a threshold of Ct 30.6. (**D**) Lesion swabs ($n = 9$) sequenced by IBL. Data are fitted with a logistic function. (**E**) Lesion swabs ($n = 27$) sequenced by the LACPHL. Data are fitted with a logistic function. (**F**) Lesion swabs ($n = 25$) from various anatomical

sites and sequenced by the MDH. Data are fitted with a logistic function. (**G**) Clinical specimens ($n = 19$) consisting of lesion swabs and crusts of healing lesions sequenced by the CEVS. Data are fitted with a logistic function. (**H**) Clinical specimens ($n = 78$) consisting of lesion swabs from various anatomical sites as well as oropharyngeal swabs sequenced by INSA. (**I**) Vesicle swabs ($n = 34$) from various anatomical sites tested in parallel on Illumina and ONT sequencing platforms by the NHS. Data from both sequencing platforms are fitted with logistic function and the dashed lines correspond to 80% genome coverage at a threshold of Ct 25.6 on the Illumina platform and Ct 24.7 on the ONT platform. (**J**) Lesion swabs ($n = 8$) sequenced on the ONT platform by JHMI. Source data can be found in S5 Data. CEVS, Centro Estadual de Vigilância em Saúde; Ct, cycle threshold; DPHL, Delaware Public Health Lab; FDH, Florida Department of Health; IBL, Idaho Bureau of Laboratories; INSA, National Institute of Health Dr. Ricardo Jorge; JHMI, Johns Hopkins Medical Institutions; LACPHL, Los Angeles County Public Health Lab; MDH, Minnesota Department of Health; NHS, National Health Service; ONT, Oxford Nanopore Technology; RIDOH RISHL, Rhode Island Department of Health/Rhode Island State Health Laboratory.

depth, which ranged between Ct 24.7 to 31.2 (**Fig 6A–6C and 6I**). For some laboratories we were not able to determine this threshold, because (1) the range in Ct values was too narrow to see the decrease at higher Ct values (**Fig 6D–6G**); or (2) there was high variation in the coverage at lower Ct values (**Fig 6H and 6J**). The reason why some samples with low Ct values failed to achieve high genome coverage is unknown, but this was likely due to technical error during library preparation or sequencing, and not due to the primer scheme itself as such patterns would have been consistent across all the laboratories. Most samples were sequenced on Illumina platforms, as this was the most commonly used sequencing platform in the laboratories that requested primers. Based on our previous experience with amplicon-based sequencing of viruses such as Zika and SARS-CoV-2, we expected that the primer scheme could also be used with Oxford Nanopore Technologies workflows. The NHS Lothian laboratory had the ability to prepare libraries with both Illumina and Oxford Nanopore technologies workflows and sequenced on the Illumina MiSeq and Oxford Nanopore Technologies GridION. This pairwise comparison showed that genome coverage at 10× read depth was comparable between both workflows (**Fig 6I**). By comparing data generated by independent public health laboratories, we show that the human monkeypox virus primer scheme can be successfully implemented in previously established Illumina and Oxford Nanopore Technologies sequencing workflows under varying conditions.

## Discussion

We developed an amplicon-based sequencing approach for human monkeypox virus to provide a more sensitive, lower cost, and higher throughput alternative to metagenomic sequencing. We used PrimalScheme to develop primers for human monkeypox virus based on a pre-outbreak A.1 lineage genome (GenBank accession: MT903345) and tested the primer scheme with clinical specimens in 2 independent laboratories. After initial validation, we shared primer pool aliquots with 10 additional public health laboratories, who successfully implemented the scheme in their existing amplicon-based sequencing workflows. Our findings showed greater breadth and depth of genome coverage when using the amplicon-based sequencing approach as compared to metagenomics, particularly for specimens with lower DNA concentrations. We identified Ct value and number of sequencing reads as 2 factors that influence percent genome coverage. Based on our findings, we made the following recommendations for amplicon-based sequencing of human monkeypox virus:

1. Utilize existing amplicon-based SARS-CoV-2 sequencing infrastructure.

2. Prioritize samples with a Ct value <31, if resources are limited.

3. Generate at least 1 million sequencing reads per sample, if resources allow.

Importantly, our human monkeypox virus primer scheme can be used with currently implemented SARS-CoV-2 sequencing workflows and Illumina or Oxford Nanopore Technologies sequencing platforms. This will enable laboratories that are currently using amplicon-based sequencing approaches for SARS-CoV-2 to expand their portfolio by including human monkeypox virus, with minimal change to their overall sequencing and bioinformatics workflow. By collaborating with 10 public health labs across the world, we show that this approach can be independently and effectively implemented across a range of settings, experience levels, and resource demands. Notably, this approach could see widespread adoption among low- or middle-income countries (LMICs), many of which saw rapid expansion of their sequencing infrastructure during the COVID-19 pandemic [21,22]. Indeed, one of the few positive developments of the pandemic has been the implementation of standardized genomic surveillance systems in LMICs, evident by the number of SARS-CoV-2 genomes submitted to genome databases from countries that previously lacked the capability for large-scale whole-genome sequencing. By utilizing many of the same sequencing resources (e.g., reagents, equipment, and bioinformatics pipelines) developed for use with SARS-CoV-2, our approach could provide a streamlined approach for genomic surveillance to aid LMICs in capacities ranging from case confirmations to outbreak response while placing little to no additional demands on resources. In particular, the outsourcing of traditionally computationally intensive workflows via the Terra platform and the ease of updating both the primer scheme and bioinformatics pipelines with our approach removes some of the barriers typically seen with implementing genomic surveillance in low-resource settings.

The comparatively steep drop-off in genome coverage with higher PCR Ct samples seen with metagenomics reinforces a limitation of this sequencing approach found in other studies that used clinical metagenomics for pathogen sequencing [23]. This lack of sensitivity presents a potential challenge to genomic surveillance, as human monkeypox virus DNA concentrations fluctuate throughout the course of infection and across specimen types [24]. We show that high genome coverage can be achieved with different workflows, sequencing platforms, and sample types such as lesion and oropharyngeal swabs. The higher sensitivity seen in amplicon-based sequencing allows for sequencing of a larger variety of sample types across a wider range of Ct values compared to metagenomic sequencing.

Given the large number (163) of amplicons needed to span the entire human monkeypox virus genome, there was an increased potential for amplicon drop-outs and a resulting reduction in genome coverage. Despite being an inherent risk of amplicon sequencing, here we identified only 2 amplicons (75 and 118) with consistent low coverage <10×, and an additional 8 amplicons with overall lower coverage. Moreover, we show that none of the sites associated with clade- and lineage-defining mutations have consistently low coverage. The lower coverage is likely the result of lower amplification efficiency and may be improved by further optimization to achieve high PCR efficiency across all the primer sets. As a double-stranded DNA virus with genetic proofreading mechanisms, the monkeypox virus has a comparatively slower evolutionary rate than single-stranded RNA viruses [25]. Less genetic diversity over time translates to fewer differences between the reference genome used to generate the primer scheme and genomes that are associated with the 2022 outbreak (clade IIb), decreasing the risk for amplicon drop-out. As a result, the primer scheme may not have to be updated as frequently as, for example, the primer scheme used to sequence SARS-CoV-2, but iterative updates can be released if needed [14]. As our primer scheme was specifically developed for clade IIb viruses, additional primer schemes may be needed for surveillance of other virus clades, such as those present in endemic regions. While we show consistently high coverage of the clade I and clade IIa defining genome sites, as well as high coverage for the majority of sites within

sub-clade A.1, our primer scheme may not be able to generate sufficiently high coverage (>80%) of genomes belonging to these clades necessary for genomic surveillance.

The intended use for this amplicon-based sequencing approach is for public health surveillance and it is not intended as a diagnostic assay. As such, we did not evaluate cross-reactivity with other orthopoxviruses. Similarly to the genomic surveillance systems used for SARS-CoV-2, sequencing should be performed with confirmed positive cases to generate data used to guide public health responses. While we did not test this approach with other orthopoxviruses, if they were to be present in a sample and able to be partially amplified, then it would be evident in the bioinformatics analysis and could serve as an additional use case for this approach. Public health surveillance has been implemented to understand factors that may have contributed to the rapid global spread of human monkeypox virus in 2022 [26], including evidence for human adaptation driven by the APOBEC3 enzyme [12]. In outbreak settings, whole-genome sequencing can be used to detect the introduction of new lineages, identify mutations associated with phenotypic adaptations, assess transmission dynamics and intervention effectiveness, and guide clinical decision-making. For example, our method has already proven successful in identifying a 600 bp deletion that affects commonly used RT-PCR–based clinical diagnostic assays [27,28]. Furthermore, our approach can be used to detect a mutation in the VP37 gene resulting in resistance to Tecovirimat, an antiviral medication widely used in the treatment of mpox, further highlighting the value of regular genomic surveillance to detect clinically important mutations [29,30].

There were several limitations in this study. First, the large number of primers included in the scheme resulted in differences in PCR efficiency between amplicons. By down-sampling the sequencing data, we show that a higher number of sequencing reads can overcome these challenges. Further optimization by designing primers targeting other positions and changing primer concentrations may help to reduce the number of raw sequencing reads needed to reach at least 10× coverage depth across the genome. Second, although results between laboratories were consistent, we observed some variability in the relation between Ct value and genome coverage. Differences in the sample types, qPCR assays, sequencing workflows, and thermocycler calibration can likely explain this variability. Third, there were some challenges with the more variable ends of the genome that contain inverted terminal repeats. This resulted in multiple binding sites for primers spanning these regions. In addition, this can create challenges when mapping reads to the inverted terminal repeats for both Illumina and Oxford Nanopore Technologies data, which may result in the misalignment of reads. This warrants careful interpretation of sequencing results in these genome regions. Lastly, orthopoxviruses are known to have genomic rearrangements (translocations, duplications, and inversions) especially in the inverted terminal repeats [31], which may not be well identified by amplicon-based sequencing. Although we have shown the efficacy of this approach in identifying mutations that affect diagnostic assays [28], amplicon-based sequencing may not be able to identify other genomic rearrangements that may have epidemiological or clinical significance. We recommend periodic long read metagenomic sequencing to supplement large-scale amplicon-based sequencing as an additional strategy to improve genomic surveillance of human monkeypox virus.

Through this study, we have shown that amplicon-based sequencing can increase the sensitivity, breadth, and depth of human monkeypox virus genome coverage with low DNA concentration specimens. By being able to "plug" the human monkeypox virus primer scheme into existing sequencing and bioinformatics infrastructure, our approach has helped public health laboratories worldwide to quickly adapt their existing workflows in response to the global mpox outbreak.

**Table 1. Ethical oversight.**

| Remnant clinical samples | State/Country | Institutional oversight | Decision | Protocol ID |
|---|---|---|---|---|
| Lesion swabs | Brazil | Centro Estadual de Vigilância em Saúde | Waived | SES-RS N° 357/2021 and SES-RS N° 849/202 |
| Lesion swabs | Connecticut, US | Connecticut Department of Public Health | Waived (deemed public health surveillance) | N/A |
| Lesion swabs | Connecticut, US | Institutional Review Board from the Yale University Human Research Protection Program | Waived | 2000033281 |
| Lesion swabs | Delaware, US | Delaware Public Health Lab | Waived (deemed public health surveillance) | N/A |
| Lesion swabs | Florida, US | Florida Department of Health | Waived (deemed public health surveillance) | N/A |
| Swabs | Idaho, US | Idaho Bureau of Laboratories | Waived (deemed public health surveillance) | N/A |
| Swabs | Maryland, US | Johns Hopkins Medical Institutional Review Board | Approved | IRB00317591 |
| Lesion swabs | California, US | Los Angeles County Public Health Lab | Waived (deemed public health surveillance) | N/A |
| Lesion and oropharyngeal swabs | Massachusetts, US | Massachusetts Department of Public Health, Massachusetts State Public Health Laboratory | Approved | 1917413 |
| Swabs | Minnesota, US | Minnesota Department of Health | Waived (deemed public health surveillance) | N/A |
| Lesion and oropharyngeal swabs | Portugal | National Institute of Health | Waived (deemed public health surveillance) | Technical Orientation n°04/2022 |
| Swabs | United Kingdom | NHS Lothian BioResource | Approved | 20/ES/0061 |
| Dry vesicle swabs | Rhode Island, US | Rhode Island Department of Health, Rhode Island State Health Laboratory | Waived (deemed public health surveillance) | 216-RICR-30-05-1 |

## Methods

### Ethics statement

As part of this study, we sequenced remnant clinical specimens that tested presumptive positive for monkeypox virus. Ethical oversight for each institution is indicated in **Table 1**. All data were de-identified prior to sharing and sample codes as included in the manuscript are not known outside the research groups.

### Primer scheme design

The human monkeypox primer scheme (v1) was designed with PrimalScheme [15] using a pre-outbreak clade IIb reference genome (GenBank accession: MT903345) belonging to the A.1 lineage, following the newly proposed monkeypox virus naming system [8]. The primer scheme comprises a total of 163 primer pairs with an amplicon length ranging between 1,597 and 2,497 bp (average length of 1,977 bp; **S1 Table**).

### Clinical specimens for validation

Validation of the primer scheme was done by MASPHL, CDPH, and YSPH. At CDPH, a total of 22 clinical specimens consisting of swabs from lesions of 12 individuals were tested. DNA was extracted from clinical specimens via the Roche MagNA pure 96 kit or manual extraction and tested with the non-variola *Orthopox* real-time PCR assay on the Bio-Rad CFX96 instrument [32].

At MASPHL, a total of 133 clinical specimens consisting of both lesion swabs and oropharyngeal swabs with and without detection of lesions in the throat were tested. DNA was extracted from clinical specimens using the QIAamp DSP DNA Blood Mini kit and tested with the non-variola *Orthopox* real-time PCR assay on the ABI 7500 Fast Real Time PCR instrument [32].

## Metagenomic sequencing

Initial validation was done by sequencing 10 clinical specimens with both metagenomic and amplicon-based sequencing approaches at MASPHL. For metagenomic sequencing, samples were first quantified using the Qubit 1x dsDNA High Sensitivity kit to determine concentration. Initial loading volume was calculated for each sample to fall in the recommended range of 100 to 500 ng. Each sample was then run in duplicate through the Illumina DNA Prep kit for library preparation following manufacturer's protocol. After Flex Amplification PCR, the libraries were cleaned using the protocol option for standard DNA input. Post-library purification, each sample library was run on the Tapestation D1000 to determine the average peak size. Samples were also again quantified using the Qubit dsDNA High Sensitivity kit to best normalize the samples when pooling the libraries. Samples were then pooled, denatured, and diluted to 10 pM before being spiked with 5% PhiX. The diluted pooled libraries were then loaded and run on an Illumina v.2 300 cycle MiSeq cartridge, with a target of 1.5 to 3 million reads per library. Genomes were generated using the Broad Institute's viral-pipelines workflows on Terra.bio using both the assemble_refbased and assemble_denovo workflows. The assemble_refbased workflow aligned reads against the MA001 (ON563414.3) genome for consensus generation. The assemble_denovo workflow scaffolded de novo SPAdes contigs against both the MA001 (ON563414.3) and RefSeq (NC_063383.1) references, followed by read-based polishing.

## Amplicon-based sequencing

At MASPHL, libraries were prepared for sequencing using the Illumina DNA prep kit [20]. Initial library clean-up in the Illumina DNA prep protocol was done by following recommendations for standard DNA input, but later optimization showed improved coverage when following the recommendations for small PCR amplicon input. A negative template control was included during library prep for each sequencing run. Pooled libraries were sequenced on the Illumina MiSeq (paired-end 150), with a target of 0.5 to 1 million reads per library. For the initial 10 sequenced samples, primers were trimmed and consensus genomes were generated at a minimum depth of coverage of 10× via the TheiaCoV_Illumina_PE Workflow Series on Terra.bio. TheiaCoV_Illumina_PE was originally written to enable genomic characterization of SARS-CoV-2 specimens from Illumina paired-end amplicon read data. Modifications to TheiaCoV_Illumina_PE were made to support monkeypox virus genomic characterization; these modifications accommodated the use of a monkeypox virus reference sequence and primer scheme for consensus genome assembly. These updates were made available in the TheiaCov v2.2.0 release [33]. The Terra platform allows for consistently deployed bioinformatics environments without the need for on-site computing infrastructure and high-level programming capabilities. This allows for consistent version control, assembly parameters, and upload to public repositories across a distributed sequencing data generation network.

At YSPH, libraries were prepared for sequencing using the Illumina COVIDSeq test (RUO version) [20]. A negative template control was included during library prep for each sequencing run. Pooled libraries were sequenced on the Illumina NovaSeq (paired-end 150), with on average 12 million reads per library (range: approximately 0.4 to approximately 20 million reads per library).

## Data analysis

Bioinformatic analyses were done by YSPH to unify the analysis for data generated on Illumina sequencers. BAM files containing mapped reads were shared, and mapped reads were extracted using bamToFastq or "BEDtools bamtofastq" (version 2.30.0) [34] into FASTQ files before downstream analysis. To start the analysis, we remapped the reads to the human monkeypox reference genome (GenBank accession: MT903345) using BWA-MEM (version 0.7.17-r1188) [35]. The generated BAM mapping files were sorted using SAMtools (version 1.6) [36] and then used as input to iVar (version 1.3.1) [37] to trim primer sequences from reads. The trimmed BAM files were then used to generate consensus sequences using iVar with a minimum read depth of 10. We also calculated the per-base read coverage depth using genomeCoverageBed or "BEDTools genomecov" (version 2.30.0) [34]. To calculate the percent genome coverage for each sample, we generated a pairwise whole-genome sequence alignment of each sample against the human monkeypox virus reference genome (GenBank accession: MT903345) using Nextalign (version 1.4.1) [38]. We then calculated the percentage of alignment positions, excluding ambiguous nucleotides and deletions, using BioPython (version 1.7.9) [39].

To investigate the effect of the number of sequenced reads on the percent genome coverage, we generated randomly down-sampled sequence reads for the CDPH/YSPH samples. We used the CDPH/YSPH samples because they were sequenced at sufficiently high depth to allow down-sampling at different total read counts, namely 0.5, 1, 1.5, and 2 million reads per sample. We first generated interleaved sequencing read files from the paired-end sequencing reads using seqfu (version 1.14.0) [40] and then randomly down-sampled the reads at the specified total read count thresholds using "seqtk sample" (version 1.3-r106) [41]. Similarly, we generated the consensus whole-genome sequences and calculated the percentage genome coverage as described in the previous paragraph.

All further data analysis and plotting were performed using R statistical software v4.2.0 [42] using the ggplot2 v3.3.6 [43], dplyr v1.0.9 [44], tidyr v1.2.0 [45], and cowplot v1.1.1 [46] packages. We fitted a logistic function specified as follows: $y = a / (1 + exp(-b * (x - c)))$, where a, b, and c are parameters and "x" is the PCR Ct value and "y" is the percentage genome coverage or completeness. We used the "curve_fit" function in the numpy Python package to fit the model [47]. To interpolate the Ct value corresponding to an 80% threshold Ct value, we rearranged the logistic equation to estimate "x" given the estimated parameters a, b, and c, and genome coverage or "y" of 80% at each read coverage depth. Fig 5 was created in R using the leaflet package [48]. The base layer of the map is sourced from OpenStreetMap (https://github.com/CartoDB/basemap-styles/blob/master/LICENSE.md) and Carto (https://carto.com/basemaps/) under a CC-BY creative commons license.

## Public health response

After initial validation, we made a detailed protocol publicly available and shared pooled primer aliquots with laboratories across the United States and internationally to support their public health genomic efforts [20]. Each laboratory sequenced samples by adapting their SARS-CoV-2 amplicon sequencing workflows, performed their own bioinformatics, and submitted sequencing data and consensus genomes to public repositories such as NCBI or GISAID (S3 Table). By sharing de-identified sequencing data (e.g., coded human-depleted raw reads or BAM files containing mapped reads) and metadata (e.g., sample codes, sample types, and Ct values), we ran the same bioinformatics pipeline on all data generated on Illumina platforms to determine breadth and depth of coverage for each genome under standardized conditions, as described above.

After initial dissemination of primer aliquots, Integrated DNA Technologies created pre-pooled primer aliquots, which eliminates the need to manually pool individually ordered primers. These pools were validated by YSPH and revealed similar coverage as compared to initial results with manually pooled primers (**S1 Fig**).

## Centro Estadual de Vigilância em Saúde (CEVS)

A total of 19 clinical specimens consisting of lesion swabs and the crusts of healing lesions were tested. DNA was extracted from clinical specimens using the Invitrogen PureLink Viral RNA/DNA Mini kit and tested with a clade-specific PCR assay on the Bio-Rad CFX opus 96 instrument. Libraries were prepared for sequencing using the Illumina DNA prep kit. Pooled libraries were sequenced on the Illumina MiSeq v3 (paired-end 150), with 2 million reads per library.

## Delaware Public Health Lab (DPHL)

A total of 10 clinical specimens consisting of dry swabs from lesion sites were tested. DNA was manually extracted from clinical specimens using the QIAGEN QIAamp DSP DNA Blood Mini Kit. Amplification was achieved by using the Perfecta Multiplex qPCR SuperMix Low Rox PCR assay in conjunction with CDC issued Non-variola Orthopoxvirus Real-Time PCR Primer and Probe Set on the Applied Biosystems 7500 Fast Real-Time PCR instrument. Libraries were prepared for sequencing using the Illumina DNA prep kit. Pooled libraries were sequenced on the Illumina Miseq (paired-end 150), with about 1 million to 2.5 million reads per library.

## Florida Department of Health (FDH)

A total of 6 clinical lesion swab specimens were tested. DNA was extracted from clinical specimens using the Qiagen QIAamp DSP DNA Blood Mini Kit and tested with the non-variola *Orthopox* real-time PCR assay on the BioRad T100 instrument [32]. Libraries were prepared for sequencing using the Illumina Nextera v2 kit. Pooled libraries were sequenced on the Illumina iSeq 100 v2 (paired-end 150), with 1 to 2.4 million reads per library.

## Idaho Bureau of Laboratories (IBL)

A total of 9 lesion swab clinical specimens were tested. DNA was extracted from clinical specimens using the QIAGEN QIAamp DSP DNA Blood Mini kit and tested with Perfecta Multiplex qPCR SuperMix Low Rox PCR assay in conjunction with the CDC FDA-approved Non-variola Orthopoxvirus (VAC1) assay on the Applied Biosystems 7500 Fast Dx Real-Time PCR instrument. Libraries were prepared for sequencing using the Illumina DNA prep kit. Pooled libraries were sequenced on the Illumina MiSeq, with 0.8 to 1.2 million reads per library.

## Johns Hopkins Medical Institutions (JHMI)

A total of 8 clinical lesion swab specimens were tested. DNA was extracted from clinical specimens using the bioMérieux easyMag and tested with a LDT PCR assay that adopted the primer and probe sequences of the non-variola orthopoxvirus, modified from Li and colleagues [49]. Total reaction volume for the real-time PCR was 20 μL (5 μL of template and 15 μL master mix). The master mix contained 5 μL TaqPath 1-Step RT-qPCR Master Mix (Applied Biosystems, A15299, Waltham, Massachusetts), 4 μL water, and 1 μL of each primer (10 nm) and the probe (5 nm) in addition to 1 μL of each primer (10 nm) and the probe (5 nm) for the RNAse P internal control target. Real-time PCR was performed using Prism 7500 Detection System

(Applied Biosystems) and the following cycling conditions: 1 cycle at 95.0˚C for 2 min and 40 cycles at 95.0˚C for 3 s and 60.0˚C for 31 s. Libraries were prepared for sequencing using NEB-Next ARTIC reagents for SARS-CoV-2 sequencing. Pooled libraries were sequenced on the Oxford Nanopore Technologies GridIon. Primers were trimmed and consensus genomes were generated at a minimum depth of coverage of 10× using the ARTIC bioinformatics pipeline [50].

### Los Angeles County Public Health Lab (LACPHL)

A total of 27 remnant lesion swab specimens were tested. DNA was extracted using the Qiagen EZ1&2 DNA Tissue Kit from specimens that were previously tested with the non-variola *Orthopoxvirus* real-time PCR assay on the ABI 7500 FastDx instrument following Centers for Disease Control and Prevention Laboratory Response Network protocols [32]. Libraries were prepared for sequencing using the Illumina DNA prep kit. Pooled libraries were sequenced on the Illumina MiSeq v2 (paired-end 150), with 0.5 to 1 million reads per library.

### Minnesota Department of Health (MDH)

A total of 25 clinical specimens consisting of lesion swabs from various anatomical sites were tested. DNA was extracted from clinical specimens using the QIAamp DSP DNA Blood Mini kit and tested with a non-variola *Orthopox* real-time PCR assay on the ABI 7500 Fast Real Time PCR instrument [32]. Libraries were prepared for sequencing using the Illumina DNA prep kit. Pooled libraries were sequenced on the Illumina MiSeq v2 (paired-end 250), with target of 500,000 reads per library.

### National Institute of Health Dr. Ricardo Jorge (INSA)

A total of 78 clinical specimens consisting of lesion swabs from various anatomical sites and oropharyngeal swabs were tested. DNA was extracted from clinical specimens using the Mag-MAX Viral/Pathogen Nucleic Acid Isolation kit and tested with a real-time PCR assay on the CFX Opus real-time PCR system [51–53]. Libraries were prepared for sequencing using the Illumina Nextera XT library prep kit. Pooled libraries were sequenced on the Illumina NextSeq 550, with 2 million reads per library.

### National Health Service Lothian (NHS Lothian)

A total of 34 clinical specimens consisting of vesicle swabs and swabs from various anatomical sites were tested. DNA was extracted from clinical specimens using the Biomerieux Nuclisens EasyMag kit and tested with the clade-specific real-time PCR assay on the Applied Biosystems 7500 Fast Real-Time PCR instrument [54]. Illumina libraries were prepared for sequencing using the Illumina COVIDSeq test (RUO version). Pooled libraries were sequenced on the Illumina MiSeq—Micro v2 reagent kit, with 400,000 reads per library. Primers were trimmed and consensus genomes were generated at a minimum depth of coverage of 5× using the Public Health Wales Nextflow nCoV-2019 pipeline that utilizes iVar [55]. Whole-genome PCR amplicons were also used to prepare Nanopore libraries using the Artic LoCost method [56], substituting Blunt TA Ligase with NEBNext Ultra II Ligation Mastermix for barcode ligation. Libraries were pooled on a single R9.4.1 flowcell and sequenced with a GridION (Oxford Nanopore Technologies) running live High Accuracy basecalling in MinKnow v21.11.6, aiming for 100,000 reads per library. Consensus genomes were generated at a minimum depth of coverage of 20× with the Artic field bioinformatics pipeline v1.2.1 and variants called with Nanopolish [55].

### Rhode Island Department of Health/Rhode Island State Health Laboratory (RIDOH RISHL)

A total of 56 clinical specimens consisting of dry vesicle swabs from various anatomical sites were tested. DNA was extracted from clinical specimens using the Qiagen QIAmp DSP Blood mini kit and tested with the non-variola *Orthopox* real-time PCR assay on the Applied biosystems 7500 instrument. Libraries were prepared for sequencing using the Illumina DNA prep kit. Pooled libraries were sequenced on the Illumina MiSeq (paired-end 150), with 1.2 million reads per library.

## Supporting information

**S1 Table. Human monkeypox virus primer scheme (v1).** Primer positions are determined based on alignment to the MT903345 reference genome.
(XLSX)

**S2 Table. Depth of coverage at phylogenetically informative genome sites.** Coverage at 46 clade- and lineage-defining positions [19] was determined based on 15 samples sequenced by the CDPH/YSPH with Ct <31 and down-sampled to 1 million sequencing reads. Positions are listed for the NCBI mpox reference genome (NC_063383) and the reference genome (MT903345) used in this study.
(DOCX)

**S3 Table. Source data for samples included in this study.** Listed are institute, specimen code, sample type, Ct value, sequencing platform, percent genome coverage at 10×, and accession numbers.
(XLSX)

**S1 Fig. Validation of pre-pooled primers (Yale hMPXV amplicon panel) created by Integrated DNA Technologies.** Lesion swabs (*N* = 21) were re-sequenced using the Yale hMPXV amplicon panel instead of manually pooled primers. NC = negative control. Source data can be found in S6 Data.
(TIF)

**S1 Data. Source data for Fig 1.**
(XLSX)

**S2 Data. Source data for Fig 2.**
(XLSX)

**S3 Data. Source data for Fig 3.**
(XLSX)

**S4 Data. Source data for Fig 4.**
(ZIP)

**S5 Data. Source data for Fig 6.**
(XLSX)

**S6 Data. Source data for S1 Fig.**
(XLSX)

## Acknowledgments

We thank Cornelius Roemer for help with the logistic function analysis.

## Author Contributions

**Conceptualization:** Nicholas F. G. Chen, Chrispin Chaguza, Luc Gagne, Glen R. Gallagher, Chantal B. F. Vogels.

**Formal analysis:** Nicholas F. G. Chen, Chrispin Chaguza, Luc Gagne, Matthew Doucette, Daniel J. Park, Nathan D. Grubaugh, Glen R. Gallagher, Chantal B. F. Vogels.

**Investigation:** Nicholas F. G. Chen, Chrispin Chaguza, Luc Gagne, Matthew Doucette, Sandra Smole, Erika Buzby, Joshua Hall, Stephanie Ash, Rachel Harrington, Seana Cofsky, Selina Clancy, Curtis J. Kapsak, Joel Sevinsky, Kevin Libuit, Daniel J. Park, Peera Hemarajata, Jacob M. Garrigues, Nicole M. Green, Sean Sierra-Patev, Kristin Carpenter-Azevedo, Richard C. Huard, Claire Pearson, Kutluhan Incekara, Christina Nishimura, Jian Ping Huang, Emily Gagnon, Ethan Reever, Jafar Razeq, Anthony Muyombwe, Vítor Borges, Rita Ferreira, Daniel Sobral, Silvia Duarte, Daniela Santos, Luís Vieira, João Paulo Gomes, Carly Aquino, Isabella M. Savino, Karinda Felton, Moneeb Bajwa, Nyjil Hayward, Holly Miller, Allison Naumann, Ria Allman, Neel Greer, Amary Fall, Heba H. Mostafa, Martin P. McHugh, Daniel M. Maloney, Rebecca Dewar, Juliet Kenicer, Abby Parker, Katharine Mathers, Jonathan Wild, Seb Cotton, Kate E. Templeton, George Churchwell, Philip A. Lee, Maria Pedrosa, Brenna McGruder, Sarah Schmedes, Matthew R. Plumb, Xiong Wang, Regina Bones Barcellos, Fernanda M. S. Godinho, Richard Steiner Salvato, Aimee Ceniseros, Mallery I. Breban, Nathan D. Grubaugh, Glen R. Gallagher, Chantal B. F. Vogels.

**Methodology:** Nicholas F. G. Chen, Chrispin Chaguza, Luc Gagne, Glen R. Gallagher, Chantal B. F. Vogels.

**Supervision:** Glen R. Gallagher, Chantal B. F. Vogels.

**Visualization:** Nicholas F. G. Chen.

**Writing – original draft:** Nicholas F. G. Chen, Chrispin Chaguza, Chantal B. F. Vogels.

**Writing – review & editing:** Nicholas F. G. Chen, Chrispin Chaguza, Luc Gagne, Matthew Doucette, Sandra Smole, Erika Buzby, Joshua Hall, Stephanie Ash, Rachel Harrington, Seana Cofsky, Selina Clancy, Curtis J. Kapsak, Joel Sevinsky, Kevin Libuit, Daniel J. Park, Peera Hemarajata, Jacob M. Garrigues, Nicole M. Green, Sean Sierra-Patev, Kristin Carpenter-Azevedo, Richard C. Huard, Claire Pearson, Kutluhan Incekara, Christina Nishimura, Jian Ping Huang, Emily Gagnon, Ethan Reever, Jafar Razeq, Anthony Muyombwe, Vítor Borges, Rita Ferreira, Daniel Sobral, Silvia Duarte, Daniela Santos, Luís Vieira, João Paulo Gomes, Carly Aquino, Isabella M. Savino, Karinda Felton, Moneeb Bajwa, Nyjil Hayward, Holly Miller, Allison Naumann, Ria Allman, Neel Greer, Amary Fall, Heba H. Mostafa, Martin P. McHugh, Daniel M. Maloney, Rebecca Dewar, Juliet Kenicer, Abby Parker, Katharine Mathers, Jonathan Wild, Seb Cotton, Kate E. Templeton, George Churchwell, Philip A. Lee, Maria Pedrosa, Brenna McGruder, Sarah Schmedes, Matthew R. Plumb, Xiong Wang, Regina Bones Barcellos, Fernanda M. S. Godinho, Richard Steiner Salvato, Aimee Ceniseros, Mallery I. Breban, Nathan D. Grubaugh, Glen R. Gallagher, Chantal B. F. Vogels.

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
