## [Editor Report · Decision Letter 0]

18 Jan 2023

Dear Dr. Vogels, 

Thank you for submitting your manuscript entitled "Development of an amplicon-based sequencing approach in response to the global emergence of human monkeypox virus" for consideration as a Research Article by PLOS Biology.

Your manuscript has now been evaluated by the PLOS Biology editorial staff, as well as by an academic editor with relevant expertise, and I am writing to let you know that we would like to send your submission out for external peer review.

Once your full submission is complete, your paper will undergo a series of checks in preparation for peer review. After your manuscript has passed the checks it will be sent out for review. To provide the metadata for your submission, please Login to Editorial Manager (https://www.editorialmanager.com/pbiology) within two working days, i.e. by Jan 20 2023 11:59PM.

Kind regards,

Paula

---

Senior Editor

PLOS Biology

---

## [Decision Letter · Decision Letter 1]

22 Mar 2023

Dear Dr. Vogels,

Thank you for your patience while your manuscript "Development of an amplicon-based sequencing approach in response to the global emergence of human monkeypox virus" was peer-reviewed at PLOS Biology. It has now been evaluated by the PLOS Biology editors, an Academic Editor with relevant expertise, and by several independent reviewers. 

In light of the reviews, which you will find at the end of this email, we would like to invite you to revise the work to thoroughly address the reviewers' reports.

As you will see below, the reviewers raise some issues that should be addressed before further consideration. We think that the manuscript would benefit of adding more details to explain what context this assay was designed for. The reviewers point to the possibility of cross reactivity with other orthopoxviruses that should be addressed, and their issues regarding the economics of this methodology. Reviewer #1 also highlights the lack of detection of strains or specificity. Please address all the reviewers' issues.

Given the extent of revision needed, we cannot make a decision about publication until we have seen the revised manuscript and your response to the reviewers' comments. Your revised manuscript is likely to be sent for further evaluation by all or a subset of the reviewers.

**IMPORTANT - SUBMITTING YOUR REVISION**

*Re-submission Checklist*

*Published Peer Review*

*PLOS Data Policy*

*Blot and Gel Data Policy*

Sincerely,

Paula

---

Senior Editor

PLOS Biology

REVIEWS:

Reviewer #1: Viral genomics and emerging pathogens.

Reviewer #2: Viral surveillance.

Reviewer #1: The work described in the manuscript "Development of an amplicon-based sequencing approach in response to the global emergence of human monkeypox virus" by Chen et al represents a technical advance in whole genome sequencing amplification of a poxvirus genome using highly multiplexed amplicon-based sequencing. However, its importance is very difficult to assess since the manuscript lacks sufficient evidence that the high coverage achieved with this method actually covers areas of significant variability and phylogenetically informative in the context of the current outbreak. In simple words, it does not matter really if the coverage is high, if it actually does not cover the phylogenetically informative sites.

The authors emphasize that there is an urgent need for a sensitive and broadly applicable sequencing approach, but they do not provide enough data to show that the amplicon-based sequencing approach is sensitive enough to detect the variations that ARE important for understanding the transmission of the virus. 

While the amplicon-based sequencing approach described in the manuscript is clearly an improvement over shotgun sequencing for whole genome sequencing of monkeypox virus, there is no evidence presented that shows that the information generated is useful for handling the outbreak. While real-time molecular epidemiology is a revolutionary tool for understanding the transmission of infectious diseases, it is not clear that this approach can be applied to large DNA viruses such as mpox virus in an extremely short timeline. The authors obviously have that information, since their sampling is comprehensive from multiple parts of the world. So, it is somehow strange that the manuscript does not include any evaluation of the diversity described.

Moreover, the authors do not provide any data on the accuracy and reproducibility of the method, which is crucial for evaluating its usefulness for public health surveillance efforts.

Additionally, the authors do not provide evidence of how cross-reactive this system would be with other mpox or orthopoxvirus. Is there a risk that this system is too strain-specific? The authors only discuss this problem in the context of how easy would be to replace/complement the primer set with new additions to fix drop outs. But they do not address how likely this panel would be able to detect other lineage introductions. 

One of the key considerations for implementing a strain-specific sequencing system like the one described in the manuscript in low- and middle-income countries (LMICs) or endemic areas of Africa where MPXV is circulating is the availability of resources and infrastructure. While the authors highlight the cost-effectiveness of the amplicon-based sequencing approach, it is important to consider the broader economic and technological landscape of LMICs and endemic areas. In many LMICs and endemic areas, there may be limited resources for public health interventions, let alone advanced molecular sequencing technologies. But, even if the infrastructure were available, it is important to evaluate the return of investment of generating the data. 

Whole-genome sequencing is a powerful tool that can provide a detailed understanding of the genomic features of the virus and its transmission dynamics. However, it is also a resource-intensive process that requires significant human and analytical resources. Therefore, it is important to evaluate whether the information generated by whole-genome sequencing of monkeypox virus is worth the investment of resources, especially in the context of limited resources during an outbreak response.

Furthermore, it is important to consider the context-specific nature of the MPXV outbreak. In many cases, outbreaks may be localized and may not require the same level of molecular surveillance as a pandemic situation. In these contexts, the cost-benefit analysis of implementing a strain-specific sequencing system would likely be very different than in a pandemic situation.

Since the authors are presenting their work basically as an enabling technology assessment, all of these components of the cost-benefit analysis should be evaluated and discussed, beyond the actual technical achievement of developing a clearly improved test.

Reviewer #2: This revision looks to establish an economically feasible sequencing approach for MPox during the 2022 outbreak. Kudos for policy translation in the discussion points. I have but minor points.

In the Discussion, it is recommended that periodic long-read metagenomics sequencing be used to identify potential rearrangements in MPox. However, is it possible that something in the amplicon sequencing might trigger this necessity rather than guessing (something like the S gene drop out in SARS-CoV-2)? Else, is the purpose of this method only for infrastructure and economic means? Is it a better move to invest in metagenomics infrastructure and pipelines that are flexible and plug-and-play for pathogens in the future? If so, then some discussion point of equity and LMIC should be made.

Regarding the impact of the manuscript, this is contextual. Is the amplicon coverage supposed to be used for identification and diagnostics or for phylogenetic studies? What is the primary intended utility of this assay?

Further, the issue of cross reactivity with other orthopoxviruses is not addressed and should be addressed at least in the Discussion.

How do the authors propose this method could be used in a fulminant outbreak where mutations are possible? Would a gene drop out be immediately detectable or predicted by this method? How does it compare to current metagenomics identification of mutations?!

---

## [Editor Report · Decision Letter 2]

20 Apr 2023

Dear Dr. Vogels,

Thank you for your patience while we considered your revised manuscript "Development of an amplicon-based sequencing approach in response to the global emergence of human monkeypox virus" for publication as a Research Article at PLOS Biology. This revised version of your manuscript has been evaluated by the PLOS Biology editors and the Academic Editor.

Based on our Academic Editor's assessment of your revision, we are likely to accept this manuscript for publication, provided you satisfactorily address the following data and other policy-related requests.

1. DATA POLICY:

A) Supplementary files (e.g., excel). Please ensure that all data files are uploaded as 'Supporting Information' and are invariably referred to (in the manuscript, figure legends, and the Description field when uploading your files) using the following format verbatim: S1 Data, S2 Data, etc. Multiple panels of a single or even several figures can be included as multiple sheets in one excel file that is saved using exactly the following convention: S1_Data.xlsx (using an underscore).

B) Deposition in a publicly available repository. Please also provide the accession code or a reviewer link so that we may view your data before publication. 

Regardless of the method selected, please ensure that you provide the individual numerical values that underlie the summary data displayed in the following figure panels as they are essential for readers to assess your analysis and to reproduce it: Figures 1, 2AB, 3, 4, 6ABCDEFGHIJ and Supplementary figure SF1.

2. I apologize for my previous title suggestion as the name of the disease is not correct. We suggest: "Development of an amplicon-based sequencing approach in response to the global emergence of mpox".

3. Please change to mpox throughout the manuscript when you refer to the disease.

We expect to receive your revised manuscript within two weeks. 

*Published Peer Review History*

*Press*

Sincerely,

Paula

----

Senior Editor,

pjaureguionieva@plos.org,

PLOS Biology

---

## [Editor Report · Decision Letter 3]

5 May 2023

Dear Dr. Vogels,

Thank you for the submission of your revised Research Article "Development of an amplicon-based sequencing approach in response to the global emergence of mpox" for publication in PLOS Biology. On behalf of my colleagues and the Academic Editor, Bill Sugden, I am pleased to say that we can in principle accept your manuscript for publication, provided you address any remaining formatting and reporting issues. These will be detailed in an email you should receive within 2-3 business days from our colleagues in the journal operations team; no action is required from you until then. Please note that we will not be able to formally accept your manuscript and schedule it for publication until you have completed any requested changes.

PRESS

Sincerely, 

Paula 

---

Senior Editor

PLOS Biology
